# Public Transport Infrastructure with Electromobility Elements at the Smart City Level to Support Sustainability

Gabriel Koman, Dominika Toman *, Radoslav Jankal and Silvia Krúpová *

Department of Managerial Theories, Faculty of Management Science and Informatics, University of Zilina, Univerzitna 8215/1, 010 26 Zilina, Slovakia; gabriel.koman@fri.uniza.sk (G.K.); radoslav.jankal@fri.uniza.sk (R.J.)
* Correspondence: dominika.tumova@uniza.sk (D.T.); silvia.krupova@fri.uniza.sk (S.K.)

**Abstract:** When implementing smart city elements, there are challenges in cities that need to be overcome. An analysis of global public transport infrastructure has indicated an upward trend in the integration of electric mobility solutions since 2022. The following research question characterises the problem on which the research was focused. RQ: What role do the smart city strategy and its overall planning play in promoting city sustainability via elements of electromobility in public transport? Cities are increasingly committed to promoting more sustainable urban mobility. This article discusses three areas of study: electromobility in public transport, the smart city concept, and sustainability. The novelty of this article has three parts, namely the intersection of the described areas; comparison and summarisation of best practice; and in-depth analysis of the selected city. The methodological approach includes the analysis of case studies, analysis of the selected city, sociological interviews, synthesis, and modelling. One of the main findings reveals that electromobility impacts the city's sustainability. It was also revealed that not all cities have already prepared their strategies focusing on this issue, which is unfavourable because careful planning supports the achieving sustainability in public transport. The findings were included in the design of solutions in the field of public transport infrastructure with elements of electromobility at the smart city level. Additionally, requirements for cities and recommendations for policy makers in selected areas were identified.

**Keywords:** infrastructure; public transport; smart city; electromobility; sustainability

## 1. Introduction

The field of transport is very complex, involving a diverse combination of numerous variables. There is no simple answer to the question of how to reduce traffic congestion in cities. However, implementing electromobility elements in public transport infrastructure at the smart city level for the needs of collecting multi-layered data from various sources is the first step towards finding a viable solution [1,2]. Some smart urban transport solutions also require near-real-time data availability. Examples include accident or slowdown alerts that offer alternative routes and integration with other transport infrastructure. These are dynamic speed signals or adaptive lanes using artificial intelligence and machine learning to intelligently guide vehicles [3].

At the heart of all initiatives are the data essential to understand road use patterns, timing of use and how these factors contribute to overall congestion. In parallel, it contributes to an efficient public transport infrastructure at the smart city level [4]. Reliable public transportation serves as a vital means of commuting to work or school, navigating the city comfortably, and offering affordable alternatives for individuals who choose not to drive or cannot drive. For millennials, public transportation serves as the preferred way to promote community connections and engage in digital socialisation [5]. The American Public Transportation Association (APTA) claims that public transportation is ten times safer per mile compared to other modes of transportation [5]. To promote citizen participation in public transport and reduce dependence on private vehicles, stakeholders need to gain a

comprehensive understanding of public transport and alternative options. The focus needs to be on real-time data usage. Additionally, focusing on creating a schedule strategy and placing new stops and routes is essential. Furthermore, it is also appropriate to integrate new tools such as solutions with electromobility elements in the field of micro-mobility [6].

The novelty of this article consists of three parts. The first is the study of the intersection of the areas of the electromobility concept, smart city, and sustainability. The second part is a comparison and summary of best practices from various smart cities around the world. The last part contributing to the novelty is an in-depth analysis of the selected city, which is a good representative for the introduction of electromobility infrastructure elements within the smart city concept for a large group of mainly European cities. Therefore, it is possible to appropriately transfer the original findings to the conditions of other cities.

The rest of the paper consists of an overview of the theoretical background in all three researched areas stated above. The next part is the analysis, comparison, and summarisation of best practice examples from the cities of Stockholm, Zürich, and Singapore. The following is an in-depth analysis of the city of Žilina, focusing on elements of electromobility in public transport. The next part of the article is the formulation of recommendations aimed at city managers and policymakers, which will help them in the future introduction of electromobility in public transport, as part of the smart city concept to achieve long-term sustainability. Our results and findings are put into a broader context as part of their comparison with the opinions of other world authors in the discussion. The conclusion presents a summary of the main points included in this article and draws attention to research limitations but also outlines possible directions for future research.

## 2. Theoretical Review

Following the issue studied in the article, it was necessary to define key areas. These include electromobility, the smart city concept, and sustainability. In addition to defining the main elements, research hypotheses were established in the following subchapters.

### 2.1. Electromobility

The importance of electromobility lies in its potential to address environmental challenges, increase energy efficiency, reduce costs, foster technological innovation, and contribute to a more sustainable transport system. Technology is constantly advancing, and the infrastructure is expanding. The development of sustainable urban transportation is important to combat climate change and reduce greenhouse gas emissions [7]. Therefore, electromobility is expected to play an increasingly important role in shaping the future of transportation [8]. Electromobility refers to the use of electric cars as well as e-bikes, electric motorcycles, e-buses, and e-trucks. Common features of all the vehicles listed are full or partial electric propulsion, means of energy storage, and the possibility of obtaining energy mainly from the electric power grid [9]. It can be stated that e-mobility is a complex area encompassing electric cars, other electric vehicles, and other elements of electric transport. National policies and strategies define that today (the year of 2023) transport must use green energy for its activities, make better use of modern infrastructure, and reduce negative impacts on the environment and natural resources (such as water, soil, and ecosystems) [10].

Electromobility in Public Transport

Public transport has undergone noticeable changes since its implementation in cities. These mainly concern environmental impacts [10,11]. Public transport should primarily use low-emission and zero-emission vehicles, especially those powered by electricity. The development of battery technology has led to a revolution in the range and operational capabilities of electric buses during the last decade. They have become a seemingly simple alternative to traditional electric vehicles in public transport, trams, and trolleybuses [12].

A sustainable urban public transport system meets the communication needs of city residents safely. This method should not endanger human health or the environment. It

should use renewable energy sources and be economically affordable. It aims to reduce emissions of harmful gases [13,14]. The system can function efficiently, maintain the economy and regional development, and avoid causing traffic congestion [15,16]. Public transport has significant social, economic, spatial, and environmental impacts. Therefore, it is an essential factor in the sustainability of the economy. This is not only due to the implementation of legislation at the European and national levels but also because of the increasingly widespread greening of urban agglomerations. It promotes the use of electrified transport forms [12] (European Commission, 2016). From the passengers' perspective, the fact that an electric or diesel bus will drive to a stop does not determine their decisions. The effective application of road transport incentives to public transport is currently a priority action. This is carried out as part of the implementation of sustainable mobility and development policies to increase the use of environmentally friendly forms of urban transport over individual transport [17].

Based on the information related to electromobility and a sustainable urban public transport system above, hypothesis H1 was defined as follows: by implementing electromobility elements in the city's public transport infrastructure, the sustainability of the city will be increased.

## 2.2. Smart City

Smart cities tend to invest in human and social capital, as well as traditional and modern information and communication technology (ICT) infrastructure [18]. One of the research areas within the smart city concept is the technological dimension (e.g., the utilisation of advanced technologies such as the Internet of Things (IoT), cloud computing, and sensors) [19,20]. Smart cities use ICT to improve operational efficiency, share information with the public, and provide higher-quality government services and citizen well-being [21].

The main goal of a smart city is to optimise city functions and support economic growth. At the same time, its task is to improve the quality of citizens' lives using intelligent technologies and data analysis [22]. The smart level of a city is determined using a set of characteristics, including [23]:

- Technology-based infrastructure;
- Environmental initiatives;
- Efficient and highly functional public transport;
- Confident and progressive city plans;
- People able to live and work in the city, using its resources.

The success of a smart city lies in the relationship between the public and private sectors. This is because much of the effort to create and maintain a data-driven environment is not included in the remit of local government. For example, smart surveillance cameras may need input from several companies. In addition to the technology used by the smart city, it is necessary for data analysts to evaluate the information provided by the systems to solve potential problems [20].

### Smart City Features

The smart city concept includes energy saving and environmental efficiency. An example is streetlights that darken when the roads are empty. Such smart grid technologies can improve numerous aspects of energy operation, maintenance, planning, and supply [24]. Smart city initiatives can be used to combat climate change and air pollution. Furthermore, this concept can be applied to provide activities such as sanitation, waste collection, or setting the fleet management systems. The combination of automation, machine learning, and the IoT enables the adoption of smart city technologies for various applications. For example, smart parking can help drivers find parking spaces and enable digital payment [25].

Another example of the implementation is innovative intelligent traffic management. This focuses on monitoring traffic flows and optimising traffic lights to reduce congestion. Ride-sharing services can also be managed via the smart city infrastructure [26]. In addition

to services, smart cities also enable the provision of security measures, such as monitoring high-crime areas or using sensors to enable early warning of incidents (such as floods, landslides, hurricanes, or droughts) [24]. Smart city technology can be used to improve production efficiency, urban agriculture, energy consumption, and so on. In parallel, smart cities have the function of connecting all kinds of services and providing comprehensive solutions for residents [23].

Since the analysis of the theoretical background of smart cities highlights the positives of this concept, we decided to study its application in a selected city in Slovakia. We chose the city of Žilina because we can observe the public transport situation and test the various functionalities of public transport in this chosen city. Additionally, we have access to the strategic documents of this city. We also established active communication with city representatives, whose perspectives on the selected issue provided us with practical pieces of information. Therefore, research hypothesis H2 was defined as follows: in the case of the city of Žilina, there is no strategy for the implementation of the smart city concept yet.

*2.3. Sustainability*

The sustainable development of transport is largely based on creating the possibility of finding a trade-off among three components: economic, social, and ecological perspectives [27,28]. Transport is one of the factors influencing poor air quality in cities. It is also a significant source of other pollutants. Additionally, road transport vehicles are the cause of noise surrounding roads. Among the solutions motivating citizens in urban areas to use low-emission public transport implemented in cities is the implementation of bus lanes [12]. Citizens expect a high quality of life [29]. This quality is considered the most important factor influencing the development of electromobility [30].

Efforts to improve quality of life and sustainability should be considered a prerequisite for the development of electromobility in urban public transport. It has the social dimension as a priority. The sustainability of public transport is a critical topic as cities seek to address environmental, economic, and social challenges. Public transport systems play an important role in promoting sustainability in multiple ways [31]. Public transport, when powered by electricity or alternative fuels, can significantly reduce greenhouse gas emissions compared to individual car usage. This has a positive impact on improving air quality [32].

Public transport systems are often more energy-efficient per "personnel" mile than private vehicles, especially if they incorporate energy-saving technologies and processes. Optimally planned public transport can promote more sustainable land utilisation, e.g., by reducing urban sprawl and maintaining green spaces. Public transportation can be more cost-effective for individuals compared to owning a private vehicle. Simultaneously, the development of public transport infrastructure creates jobs in construction, maintenance, and operation. Efficient public transport will alleviate traffic congestion, which can lead to economic benefits due to reduced time lost in transport and increased productivity [33].

Achieving sustainability in public transport requires careful planning, investment, and continuous efforts to address the emerging challenges. The allocation of green funds for urban planning projects should prioritise climate actions, especially in transportation [34]. The main challenges include funding, infrastructure maintenance, and the integration of new technologies. Governments, communities, and transit agencies must cooperate while addressing this issue. This is the only way to ensure that public transport systems are designed and operated in a way that maximises their sustainability benefits. This includes electrifying transit fleets, expanding service networks, and promoting the use of public transport as a convenient transport option for all [35].

Following the presented information, we defined the last hypothesis (H3). This focuses on achieving sustainability and was defined as follows: the achievement of sustainability in public transport with electromobility elements must be supported by careful planning.

To clearly display the main findings from all three researched areas, a theory analysis summary table was created (Table 1).

**Table 1.** Summary of the theoretical review.

| Authors | Key Area | Findings |
|---|---|---|
| [7,8,10,16] | Electromobility | — Important to combat climate change and reduce greenhouse gas emissions<br>— Important role in shaping the future of transportation<br>— Contributes to a reduction in harmful gas emissions<br>— Important for the implementation of sustainable mobility and development policies<br>— Helps increase the use of environmentally friendly forms of urban transport over individual transport |
| [20,24,32] | Smart city | — Tend to invest in human and social capital, traditional and modern ICT infrastructures<br>— Uses ICT to improve operational efficiency, share information with the public, provide higher quality government services and support citizen well-being<br>— Improving the quality of citizens' lives using smart technologies and data analysis<br>— Ride-sharing services are managed via the smart city infrastructure<br>— Connecting all kinds of services and providing comprehensive solutions for citizens |
| [31,33,35] | Sustainability | — The sustainability of public transport is a critical topic as cities and communities seek to address environmental, economic, and social challenges<br>— Public transport systems play an important role in promoting sustainability<br>— Public transport, when powered by electricity or alternative fuels, can reduce greenhouse gas emissions compared to personal car usage<br>— Efficient public transport will alleviate traffic congestion, which can lead to economic benefits due to reduced time lost in transport and increased productivity<br>— Challenges include funding, infrastructure maintenance, as well as the integration of new technologies |

The knowledge gap stems from the fact that compared and summarised studies by world authors mostly focus on only one of the three selected areas. Although they point to the many benefits of electromobility in public transport for achieving the sustainability of smart cities, they do not pay enough attention to the strategy and planning considering this topic. The literature also does not provide sufficiently described specific points applicable in cities of comparable size to the city of Žilina.

## 3. Materials and Methods

The studied problem was approached via the following research question RQ: what role do the smart city strategy and its overall planning play in supporting the city's sustainability via elements of electromobility in public transport?

The main objective of the article is to propose recommendations for improving public transport infrastructure in the city of Žilina and similar cities. This should be performed based on the analysis of theoretical knowledge and other analyses performed on the selected data. Additionally, the focus needs to be on the elements of electromobility within the framework of smart city standards to support sustainability.

Various methods and techniques were applied to fit the content of the chapters. The methods are divided into four areas: methods of obtaining data, methods of processing the data, methods of problem-solving, and methods of evaluating the proposed solution.

The method of document analysis was used to obtain relevant data. The selection of analysed documents was subject to a thorough process, with deliberately set criteria.

For the selected documents to meet the level of the analysis' needs, they had to meet the defined criteria:

- Area focused on public transport elements at the smart city level;
- The author of the publication is an expert in the field of issues studied.

Subsequently, to include the case study and the examined documents in the target group, they had to meet conditions such as:

- The city meets the general smart city characteristics;
- The city is positively evaluated regarding mobility;
- The city is in the selected smart cities ranking.

To effectively ensure the acquisition of relevant data from the examined documents, the following steps were defined:

1. Determination of the selected case's relevance, based on predefined criteria;
2. Description of the selected case study containing basic characteristics of the city;
3. Justification for the implementation of electromobility elements in public transport at the smart city level;
4. Indication of positive benefits for the city and examples of impacts on the city management;
5. Comprehensive evaluation of the case study containing a generalisation of the lessons learned.

The interview method was used to obtain primary information from a relevant representative of the selected city. For this method, the mayor was approached. In the structured interview, he was asked questions from the studied area and his attitude to the selected issue was observed.

The method of qualitative evaluation was applied in the synthesis of conclusions obtained from analyses conducted. This method represents an expert estimation. The comparative method, applied in the theoretical and analytical part, allowed the comparison of theoretical findings with the analytical results while enabling the comparison of opinions gained from the interview with the actual situation observed in the city.

The modelling method was used to design the solution for improving public transport infrastructure in the city at the smart city level with electromobility elements. This proposal aims at promoting sustainability. The method of logical reasoning was applied while designing the proposals and evaluating them. The method of induction was applied to obtain general conclusions for the analytical and theoretical parts.

Deduction was applied to form original perspectives via general claims. The synthesis represents the combination of findings into a unified conclusion, with the identification of connections among the aspects studied. Conclusions were drawn to extract the overall picture of the area studied. The proposed methodology is supported by studies with similar procedures applied [36–39].

An important part of the methodological approach is the operationalisation of hypotheses that were set in this research. This is carried out using the indicators listed in the following table (Table 2).

Setting the specific indicators enabled the application of this structure in all parts of the analysis performed. Focusing on the indicators leads to the possibility of testing the hypotheses' validity.

**Table 2.** Research hypotheses' operationalisation.

| Research Question (RQ) | Hypotheses | Indicators |
| --- | --- | --- |
| What role do the smart city strategy and its overall planning play in supporting the city's sustainability via elements of electromobility in public transport? | H1: By implementing electromobility elements in the city's public transport infrastructure, the sustainability of the city will be increased. | electromobility elements; public transport infrastructure; sustainability |
| | H2: In the case of the city of Žilina, there is no strategy for the implementation of the smart city concept yet. | long-term goals; strategy for smart city |
| | H3: The achievement of sustainability in public transport with electromobility elements must be supported by careful planning. | electromobility elements; specific plans |

## 4. Results

The results presented in this article have been divided into separate sections according to the analysis from which they were obtained. These are (1) the analysis of the case studies and documents, (2) the analysis of the selected city, and (3) the interview with the mayor of the selected city. The last part is a summary and interpretation of the results within (4) a set of proposed solutions for the selected city.

The following scheme (Figure 1) represents a graphic structure of individual parts included in the results.

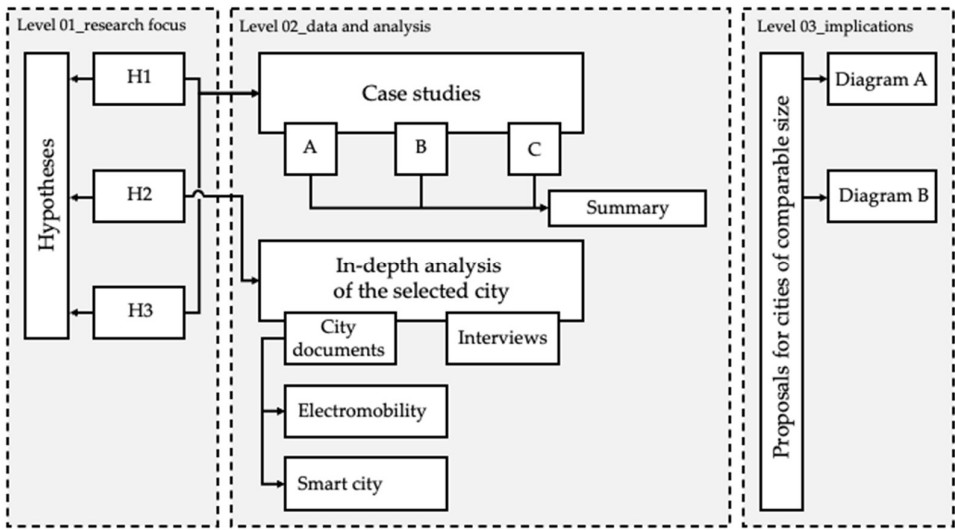

**Figure 1.** Scheme of the research levels.

These were covered under three levels: (01) research focus, (02) data and analysis, and (03) implications. The first level represents the definition of three hypotheses, which result from the analysis of theoretical background and are closely connected to the second level. Hypotheses H1 and H3 follow the analysis of case studies and H3 is connected to the in-depth analysis. As part of the case study analysis, three studies were examined in detail, followed by a summary. The second supporting part of level 02 was an in-depth analysis focused on the selected city. Here, selected documents (related to electromobility and smart city infrastructure elements) were examined. Additionally, interviews were conducted with

the city representative. The last level captured in the scheme is the implications. Specific proposals are designed in the article using two detailed diagrams.

*4.1. Case Studies' Analysis*

The analysis of secondary sources in the article included the method of case study analysis and document analysis. This primarily involves qualitative examination and evaluation of data from the documents. The main objective of the analysis was to identify the possibilities of using the elements of electromobility in public transport at the smart city level. To fulfil this objective, the analysis process focused on these elements at the selected level.

4.1.1. Case Study A—Stockholm

The level of Stockholm's relevance at the smart city level with electromobility elements was identified using the predefined criteria. The condition defining the characteristics of a smart city was met first. The smart city strategy was formally adopted by the Stockholm City Council in 2017. The second criterion was a positive assessment of mobility. Sustainable urban mobility is a priority for the Swedish capital city. Due to a holistic long-term approach and clearly defined targets, the city managed to achieve the value of 100% renewable energy for the metro, buses, and local trains in 2017. The city also managed to reduce emissions by 75% compared to 2009 levels and consume 15% less energy per passenger kilometre than in 2011 [40]. The third criterion is the city's position in the selected smart city ranking. Based on its performance, the City of Stockholm is ranked 25th by the International Institute for Management Development (IMD) [41].

Stockholm City Description

Stockholm is famous for its location among the island's waterways. Stockholm has 100 metro stations and a free network of trams, buses, light rail, and commuter trains. What makes Stockholm's transit system positively rated is its intermodal functionality using the same fare card. The main achievement of the public transit system in Stockholm is the high degree of reliability, frequency, and intermodal connectivity. Commuter rail lines connect with trams and buses at almost seamless interchange points. This enables short downtimes between modes [42]. Waiting is minimised by the high frequency and reliability of rail and bus modes. Stations and bus stops contain timetables and maps. Another positively rated aspect of Stockholm's overall transport system is the connectivity of the city via well-placed cycling and walking paths [43].

Rationale for the Implementation of Elements for the Improvement of Public Transport Infrastructure at the Smart City Level with Electromobility Elements in the City of Stockholm

Urban micro-mobility is built on shared e-bikes and e-scooters. The popularity of e-scooters has posed a challenge for the city in terms of their parking. There are currently seven operators in the city with a total of over 20,000 vehicles, so the streets were flooded with shared e-scooters. This posed an obstacle to pedestrians, cyclists, and other road users. To alleviate this challenge, more than 100 parking stands are now distributed in Stockholm, creating more order and space for micro-mobility, contributing to a greener city. The initiative is funded by Tiér and e-scooter operator Voi and is implemented in close cooperation with the City of Stockholm [44]. The City of Stockholm must create an urban environment where e-scooters become a natural part of a sustainable way of moving around the city. One of the big challenges has been the way how users park the e-scooters.

Benefits of Implementing Elements for Improving Public Transport Infrastructure at the Smart City Level with Electromobility Elements in the City of Stockholm

In the new e-scooter parking stands in the City of Stockholm, it is possible to place ten e-scooters per rack. The stands are open to all, which is one of the success factors. The aim of this project is to make the racks a starting point for a reduction in the number

of incorrectly parked e-scooters. Together with the Nordic Micro Mobility Association (NMA), those involved in the initiative believe that this will contribute to a more sustainable and accessible urban environment. Research by the Norwegian Institute for Transport Economics has shown that dedicated parking areas can improve the orderly parking of e-scooters in cities. Some of the key findings are that parking stands located in areas with a high frequency of scooter rentals provide a better effect [45].

According to the Swedish Transport Agency, improved cycling infrastructure is one of the most important measures to prevent e-scooter accidents. Dedicated parking space for shared micro-mobility is a critical component of enhanced cycling infrastructure in line with the city's green transition [46]. A study conducted in 2020 showed that parking racks for e-scooters are an important element for increased use of public transit. The placement of parking spaces at railway stations and the integration of public transport with micro-mobility services increased train travel by 35% [47]. This shows that parking stands as mobility nodes contribute to the increase in multimodal transit with public transport. This is how they help decrease the need for cars in cities. From a long-term perspective, the green transition could benefit from the reallocation of spaces now reserved for car parking among the spaces for micro-mobility.

Evaluation of the Stockholm City Case Study

As part of the City of Stockholm's analysis, the 2021 Car Dependency Report was examined. The International Transport Forum supports the recommendation for car parking spaces to be reallocated for micro-mobility, including electric scooters, electric bikes, and bicycles [46].

Another key element is the cooperation of the participating entities. Stockholm invites other actors (municipalities and authorities) to participate and take responsibility for a better micro-mobility infrastructure in Stockholm. This will ensure the promotion of environmental friendliness and sustainability. Cooperation is also important for safety and the implications for the global micro-mobility industry.

4.1.2. Case Study B—Zürich

The relevance of Zürich at the smart city level with electromobility elements was identified using the three predefined criteria. For the City of Zürich, "smart" means connecting people, organisations, and infrastructure to create social, environmental, and economic added value. The networking of data, sensors, and applications enables more efficient solutions for users of the infrastructure as well as for those operating it. A good connection with the people strengthens the possibilities for participation and contact with the administration. Based on the indicators, the city meets the first criterion of the smart city characteristics [48].

The city of Zürich emphasises mobility by making public transport attractive to users via the intermodal application "Zürimobil". This provides real-time traffic information as well as alternatives for walking, cycling, and other types of mobility. The second criterion, a positive mobility rating, is fulfilled based on the following information. Zürich is ranked 2nd according to the smart city index for 2021. The criterion on the ranking in the smart city index is thus met [49].

Zürich City Description

In the "Zürich Strategies 2035", the City of Zürich has defined areas of action with specific perspectives. Various projects are already using digital implementation options based on the city's IT infrastructure [50]. The city council has three strategic priorities within the smart city area. One of them is a form of integrated public mobility, which covers electric mobility, digitalisation, and multimodal applications [51].

The public mobility offer of Zürich is to be expanded in a resource-saving way and made available to users. For example, by replacing diesel buses with trolleybuses or electric buses, thus creating a mobility platform for the Zürich City region [50]. The city also

calls for the active participation of the citizens. They have central access to participate in city projects and processes. In the meantime, opportunities for participation are being developed and new forms of digital participation are being introduced. The processes and performance of the implemented elements are regularly reviewed, evaluated, and adapted if necessary [50].

Rationale for the Implementation of Elements for the Improvement of Public Transport Infrastructure at the Smart City Level with Electromobility Elements in the City of Zürich

The City of Zürich has prioritised smooth and efficient public transport infrastructure, sustainability, and digitisation. Mobility and transport in an urban environment consist of online city management information and strong population growth without spatial expansion. The City of Zürich has been pursuing a consistent transport policy for more than 20 years with the following effects: the share of green modes of transport is increasing, more people are travelling multimodally, and the city's initiatives have defined a target to reduce motorised private transport by 33% [52].

The City Council's goal was and still is to actively promote multimodality for flexible mobility with public transport. The city, in collaboration with transport companies and other stakeholders, launched a pilot multimodal transport project. It aimed to make public transport more attractive, reduce $CO_2$ emissions from private cars, and enable smooth and efficient transport. The key figures for the pilot project were the Zürimobil Strategic Innovation Project Zürich, integrated public mobility, the smart city strategy, and Strategy 2035 [50].

Benefits of Implementing Elements for Improving Public Transport Infrastructure at the Smart City Level with Electromobility Elements in the City of Zürich

In the form of a multimodal application, the project is a simple, user-friendly, information-providing, planning, and booking platform for local mobility services and transport hubs in the City of Zürich. Via the application, the project has combined all forms of public transport and alternative mobilities as an interface for the citizens into one product. The platform serves as a key building block for the city to make efficient use of digitisation and automation [53].

Zürimobil enables the digital pooling of transport options in Zürich. The application also provides travel information, planning, and booking for all modes of transport. It integrates mobility-related services. The City of Zürich developed the application with user interaction, effectively bringing it closer to users and meeting their user requirements. The application has a simple "user-friendly" interface. It is also flexible. With the Zürimobil application, passengers can find their current best connection and the right means of transport for each destination. The selected connection can be compared and selected according to the criteria of the journey's length, distance, and price [53].

Evaluation of the Zürich City Case Study

The application platform involves the processing of big data, so the city had to determine how that data would be used and analysed. At the same time, they focused on visualisation on a unified platform, which was pivotal for users [54]. For the city, the application is a flexible tool to track citizens' mobility, activities, and preferences. Based on the data, the city can streamline the mobility options that are most in demand. The city can efficiently decide on the next bike share station, where to place e-scooter racks, and which connections need to be reinforced. Additionally, the city can plan stops for charging e-buses based on data from the application [53]. This way, the city emphasises cooperation and collaboration among partners and developers, which is also essential. The city is taking an agile approach not only on a technical level but also in terms of partnerships.

4.1.3. Case Study C—Singapore

The relevance of Singapore City as a smart city with electromobility elements was identified based on three predefined criteria. In 2014, Singapore launched the Smart Nation Initiatives to leverage the technologies introduced by the 4th industrial revolution and

implement them on a national scale [55,56] (Smart Nation, 2021; Singapore Government Agency, 2022). The city meets the first criterion of the smart city characteristics.

There are ongoing projects in Singapore to reduce traffic congestion and improve road safety by combining artificial intelligence with investment in sustainable infrastructure [57]. Based on the information above, the criterion for a positive mobility score is met. Singapore is recognised as one of the top three smart cities in the world. It is constantly experimenting with aspects of city development that are built on the principles of sustainability and innovation. Singapore was the first to be awarded the smart city title [58].

Singapore City Description

As a land-poor city with a growing population, Singapore needs efficient transport infrastructure to function optimally. Therefore, Singapore identified smart urban mobility as one of its strategic projects [59].

Rationale for the Implementation of Elements for the Improvement of Public Transport Infrastructure at the Smart City Level with Electromobility Elements in the City of Singapore

The utilisation of digital technology is crucial to finding smart solutions that enhance the implemented public transport system, provide greater convenience and reliability, and support Singapore's vision.

Another dominant feature is the MyTransport.SG mobile application, which was launched in 2011, a service being developed by the Land Transport Authority (LTA) of Singapore. This application strives to be an all-in-one mobile application that provides useful pieces of information and features for local travellers to easily navigate the available transport devices. It provides real-time local traffic alerts, live footage from traffic cameras, and the ability to report road accidents [58,60]. While innovating the application, Singapore wanted to include a multimodal aspect to convey the ability for users to find connections where they could travel by different means of public transport or on foot [61].

Benefits of Implementing Elements for Improving Public Transport Infrastructure at the Smart City Level with Electromobility Elements in the City of Singapore

New smartphone applications have enabled Singaporeans to plan journeys with a mix of transport modes that best suit their needs. In 2018, to further enhance its travel experience, the LTA enhanced the MyTransport.SG application in Singapore with a multimodal journey planner that provides customised real-time information on different transport modes [61].

By analysing anonymised data collected from commuters' tickets, LTA can easily identify frequent points for commuters. This helps manage the fleet more efficiently and implement micro-mobility features such as charging racks and e-bike stations. The LTA has seen improvements in areas of public transport, specifically a 92% reduction in the number of overcrowded bus services despite a year-on-year increase in average daily passenger numbers, and a seven-minute reduction in average waiting times for busy bus services [62,63].

Evaluation of the Singapore City Case Study

Singapore offers a highly integrated and sophisticated transport system that achieves its purpose of providing an affordable, sustainable, and well-organised mass transport system [57]. Although some policies in Singapore may not be popular for individuals, such as the high cost of car ownership, the benefits of public transport outweigh them. For example, this is the advantage of providing efficient and innovative public transport for the larger group of people, which is successful in reducing congestion and ensuring that Singaporeans can successfully move around their city [64]. The public is positive about the current mobility situation [57]. More than 80% of the residents surveyed were satisfied with the overall public transport setup. Furthermore, citizens were satisfied with thirteen

out of the fourteen aspects analysed, including e-services, e-bike stations, trip planners, and others [65].

### 4.1.4. Conclusion from the Case Studies' Analysis

Cities and states prioritise public transport and related aspects such as improved quality of life for residents, lower levels of air pollution, and the impact on the overall environment in the surrounding area. Promoting public transport at the smart city level with electro-mobility features makes travelling easier and reduces congestion, leading to a less polluted, cleaner, and healthier environment. The smart city aspect is being promoted globally, creating innovative opportunities for cities. Based on the case studies analysed, it can be stated that smart cities should integrate ICT into the public transport infrastructure to improve efficiency and service performance while reducing pollution and speeding up mobility in the city. A summary of the results from the case studies is shown in Table 3.

**Table 3.** Summary of the case studies.

| Case Study | Rationale | Benefits | Requirements |
|---|---|---|---|
| A | 20,000 + e-scooters | Protecting vulnerable road users | Reservation of space for e-scooter racks |
| | | Parking space available within a minute's walk from the rider's destination | Creating a plan for the implementation of parking racks |
| | Increasing multimodal travel by public transport | Parking spaces at railway stations increase train travel by 35% | Cooperation with stakeholders |
| B | Priority for a seamless and efficient public transport infrastructure Residents' participation | ZüriMobil application Reducing greenhouse gas emissions by 50% by 2025 Reducing motorised private transport in municipal transport by 33% | Digitisation of public transport vehicles Sensor implementation strategy for data capture Cooperation with stakeholders |
| C | Uncomfortable travel | A 92% reduction in the number of overcrowded bus connections | Efficient data management |
| | Citizens' dissatisfaction with waiting times | Seven-minute reduction in average waiting time on busy bus services | Optimising data analysis |
| | Lack of places to park e-bikes | Implementation of racks for e-bikes | Cooperation with stakeholders |

Councils want to simplify the use of public transport, introduce residents to multimodal travel options and make public transport more accessible to all. Furthermore, they are also considering other transport users. Cities are aware that infrastructure is crucial when planning the implementation of electromobility elements at the smart city level. However, the change will not come overnight. It requires holistic strategic plans that gradually achieve the goals over the years.

Promoting public transport while using connected technologies to optimise traffic flow is a key task for the city. Reducing passenger waiting times and keeping traffic moving efficiently are two of the main objectives of public transport at the smart city level with electromobility elements.

Based on the evaluation of the case studies, it was possible to test two defined hypotheses. A sustainable public transport system is a transport system that meets the communication needs of urban citizens safely. It does not endanger human health or the environment, uses renewable energy sources, and is economically affordable for citizens while reducing harmful gas emissions. Elements of electromobility are the guarantor of ensuring emission-free mobility in 2023. Additionally, the analysis of the case studies showed that the implementation of the electromobility elements identified a decrease in carbon emissions while increasing the sustainability of public transport. Considering these results obtained from the analyses, it is possible to confirm H1: the implementation of elec-

tromobility elements in the city's public transport infrastructure increases the sustainability of the city.

The analyses highlighted the importance of the strategy in the context of a sustainable smart city. This lies in its key role in guiding and coordinating efforts to achieve long-term environmental, social, and economic goals. A well-defined and well-prepared public transport strategy with electromobility elements provides a roadmap for the effective integration of smart technologies, urban planning, and resource management, contributing to the overall sustainability of the city. The results confirm that such a strategy can be defined as an aid for optimised resource utilisation and for the increase in resilience. The strategy builds on the challenges of energy efficiency, transport, and environmental protection, promoting a holistic and sustainable approach to the development of a sustainable smart city. Linking these results to the defined hypothesis H3: achieving sustainability in public transport with electromobility elements must be supported by careful planning; it can be stated that H3 was confirmed. The main results used to test the validity of the hypotheses, following the indicators set in the methodology, are listed in Table 4.

**Table 4.** Testing validity of H1 and H3.

| Hypotheses | Indicators | Results Connected to Indicators |
|---|---|---|
| H1: by implementing electromobility elements in the city's public transport infrastructure, the sustainability of the city will be increased. | electromobility elements | electric means of public transport (e-buses, trains), electric micro-mobility (e-scooters, e-bikes) |
| | public transport infrastructure | seamless connection between modes of transport, traffic cameras, e-scooters parking stands, supporting IT infrastructure, sensors, applications, digitisation, big data, increased usage of trains, cycling infrastructure, smart solutions, AI, multimodal transport, hubs |
| | sustainability | sustainable micro-mobility, sustainability based on cooperation of entities, respecting sustainable principles |
| H3: the achievement of sustainability in public transport with electromobility elements must be supported by careful planning. | electromobility elements | electric means of public transport (e-buses, trains), electric micro-mobility (e-scooters, e-bikes) |
| | specific plans | goals set, specific strategies and their relevant parts, priorities set, transport policy |

*4.2. In-Depth Analysis of the Selected City*

There are more than 800 cities in Europe, within its 44 countries. Of these, 312 have between 50,000 and 100,000 residents, according to the 2020 census, representing approximately 24 million citizens altogether [66,67]. Žilina is a regional and district city and, with a population of around 100,000, ranks among the described 312 cities [68]. This article serves as a model example for the development of proposals for the improvement of public transport infrastructure at the smart city level with elements of electromobility. Žilina is also an administrative, transport, economic, and cultural centre for the citizens of north-western Slovakia.

4.2.1. Analysis of the City's Infrastructure in Terms of the Existing Level of Support for Electromobility

In Žilina, vehicles such as trolleybuses and electric buses represent aspects of electromobility in public transport. Žilina has two charging stations for electric bicycles, located

directly at the Žilina City Hall. The public moves around Žilina by public transport, using micro-mobility such as electric scooters [69]. The number of people transported by public transport in 2021 was 16,805,807 [70]. Residents have the possibility to use Wi-Fi in public transport vehicles; some also have a USB cable adapter, and from 2022, they can pay for the ticket directly on the bus with a payment card, mobile phone, watch, or transport card of the Transport Company of the City of Žilina (DPMŽ).

Another form of mobility provided by the city of Žilina and KIA is represented by shared bicycles. Bicycles are available to citizens from spring to autumn months [71]. A negative aspect is the lack of transport and the return of bicycles to busy places in the city by the provider. In the summer months, it is common for all available bicycles to end up at the Žilina Waterworks, where people use them for recreation and then do not return them to the city centre. Based on this observation, it can be concluded that people use other alternatives of transport such as buses, or car sharing such as Bolt, or they will not return to the city centre. The bicycles are therefore gathered in one place and are missing in the city centre. The bikes are not transported back to the city centre until the next morning.

### 4.2.2. Analysis of the City Infrastructure in Terms of the Existing Smart City Level

To analyse the infrastructure supported by smart city elements in Žilina, specific projects were selected and examined in detail.

#### Smart Traffic Lights—Preference for Public Transport

In cooperation with DPMŽ, the city launched a comprehensive system of active public transport preference. The system aims at reducing delays in transport connections and savings on fuel. The main incentive for the project is the preference of the public transport fleet at traffic lights and enabling compliance with the public transport timetable. Emphasis is placed on the demands of the citizens. Their expectations are set on the exact departure and arrival times of public transport vehicles [72] Žilina benefited from funding from the Integrated Regional Operational Programme, with total costs amounting to EUR 1.955 million. The project was co-funded by the city of Žilina with a 5% share, the state budget with 10%, and EU funds with 85% [73].

#### The SOLEZ Project Implemented

The city of Žilina was involved in the SOLEZ project. This focused on reducing carbon emissions. The project aimed to facilitate the international sharing of experience and motivate participants to use modern technologies for low-emission mobility solutions. The interfaces of the project were the study regarding the possibility of deploying electric buses in the operation of DPMŽ and the implementation of measures for parking options in the city centre using intelligent systems. The costs associated with the implementation of the project amounted to around EUR 105,000, of which approximately EUR 72,000 was covered by the Interreg Central Europe Programme [74].

#### Cooperation with the University of Žilina

The city of Žilina effectively uses the strategic location of the University of Žilina and cooperates with the academic area on various projects. In the field of mobility, the city has created the Žilina Territorial Transport Master Plan in cooperation with the University of Žilina. They also participate in traffic monitoring. At the same time, the city uses the university as an advisory body in the implementation of projects, considering ecological, mobility, and public space aspects.

#### Smart City Platform City Dashboard

The city of Žilina has a platform from the company Invipo [75]. This platform connects data from different technologies and systems into a single unit and offers clear tracking of outputs and efficient management of smart projects [76]. The portal is also available for smartphones via the Smart Žilina application. Additionally, real-time tracking of buses

and trolleybuses is possible, allowing the passenger to increase the overview of the public transport in the city.

### 4.2.3. Conclusions from the Analysis of the Selected City

The city of Žilina is on its way to reaching the smart city level, which is attractive for its residents. The projects help the city create a certain level of smart city concept. There were 431 elements of public transport infrastructure at the smart city level with elements of electromobility in the city of Žilina in 2023. The individual elements with their number are listed in the Table 5.

**Table 5.** Smart-city-level infrastructure elements with electromobility elements—city of Žilina.

| Description of the Element | Number |
|---|---|
| Cameras at junctions | 9 |
| Tracking devices on buses and trolleybuses | 88 |
| Cameras to monitor traffic in the city | 10 |
| E-bikes | 145 |
| E-bike stations | 30 |
| In total | 282 |

These elements were identified from the analyses performed and their numerical evaluation was complemented to make further comparisons.

### 4.3. Sociological Inquiry on the Topic of Smart City and Electromobility

To obtain more in-depth internal information about the city of Žilina and the city's attitudes towards the smart city concept, a structured interview was conducted with the mayor of the city, Mgr. Petr Fiabáne. The questions were sent to the mayor online in advance. The mayor had the opportunity to prepare the answers.

### 4.3.1. Free Transcript of the Interview with the Mayor

Based on the interview with the representative of the city of Žilina, the following findings emerged. The mayor is familiar with the smart city concept, participates in smart city conferences, and communicates with representatives of smart cities in Slovakia. The smart city concept is perceived by the city as an innovative solution, oriented towards the citizens and businesses operating in the city. Management support, decision support, real-time data, and smart technologies adapted to the citizens are considered among the most important points.

According to the mayor, the overall smart city concept should serve to improve and simplify the lives of residents and visitors of the city. Thus, the main goal is to improve the quality of people's lives, meeting their expectations and needs at the smart city level. The city wants to focus on the environment, e.g., waste management, as well as on transport and mobility, monitoring, and air quality. These elements should be part of a strategy that explicitly defines the direction the city should take and the specific activities to be implemented. The mayor is open to innovative and optimal solutions, but these will have to be analysed in advance because he was concerned about inefficient solutions.

### 4.3.2. Conclusion from the Interview with the Mayor

The main priorities arising from the interview with the mayor in the field of public transport are primarily oriented towards the informatisation of public transport. The city of Žilina has a form of smart city, but it does not contain all the necessary links and information and neither the citizens nor the city can make effective decisions based on this platform.

Orientation towards public transport and alternative forms of transport is a priority for the city in terms of relieving the city of the high number of cars. The city wants to provide citizens with a smooth traffic situation without the congestion that often occurs.

In addition, the city is open to solutions with elements of electromobility because of the positive effects on the noise level and the level of emissions. The city's strategic projects require efficient and consistent management. There is a need to change the urban planning principles. When planning suburban streets and urban spaces, the comfort of driving a private car should not be a priority. The city of Žilina stated that there is no systematic approach in the city administration, which is not aligned with the sustainable concept. However, the situation creates room for improvement. In 2022, the city of Žilina was preparing the creation of a coordinator position for the smart city concept. Their role would consist of intensive work on the smart city concept, effective coordination of projects and cooperation with the private sector.

The city of Žilina has the ambition to become a smart city and to support the implementation of electromobility elements into the public transport infrastructure. The city is also involved in several projects that have electromobility elements. Nevertheless, the results of the analyses identified that the city does not have a unified strategy for the implementation of the smart city concept yet, thus confirming H2: in the case of the city of Žilina, there is no strategy for the implementation of the smart city concept yet. Table 6 summarises the main results connected to the indicators needed for the testing of H2, supporting the outcome above.

**Table 6.** Testing validity of H2.

| Hypotheses | Indicators | Results Connected to Indicators |
|---|---|---|
| H2: in the case of the city of Žilina, there is no strategy for the implementation of the smart city concept yet. | Long-term goals | The official goals for this area not established yet |
| | Strategy for smart city | Absence of a detailed and approved strategy at the moment |

### 4.4. Proposals for Supporting the Implementation of the Smart City Concept in Žilina

Based on the analyses performed, a solution for the selected city of Žilina was defined to increase the number of passengers using public transport and alternative forms of transport by 10%; 10% was determined for the possible procurement and commissioning of the proposed solutions. At the same time, this goal was discussed with the representatives of the city and was set to be fulfilled within three years.

#### 4.4.1. Adding the Possibility to Search for Multimodal Transport to the Mobile Application

A suggestion is to implement the possibility of searching for multimodal transport into the Smart Žilina mobile application. Based on the analyses, it was revealed that smart platforms represent a dominant element in smart cities. City data, along with analytics and machine learning, improve the engagement and inclusion of its residents and visitors. Stakeholders such as local governments, urban service providers, industry, and residents benefit from the development of smart and well-connected urban digital platforms. The city of Žilina has a smart city platform from Invipo, implemented in 2023. In further development, the platform should enable online monitoring of traffic flows and interoperability. These elements will contribute to increasing the efficiency of process management in transport and in other areas as well. Residents and visitors could also effectively participate in the platform, having real-time access to selected modules. The Smart Mobile App for Multimodal Transport is a platform that offers local authorities the tools to analyse, develop, and implement a smart transport management system. To describe the dynamic aspects of the application system, an activity diagram was used (Figure 2).

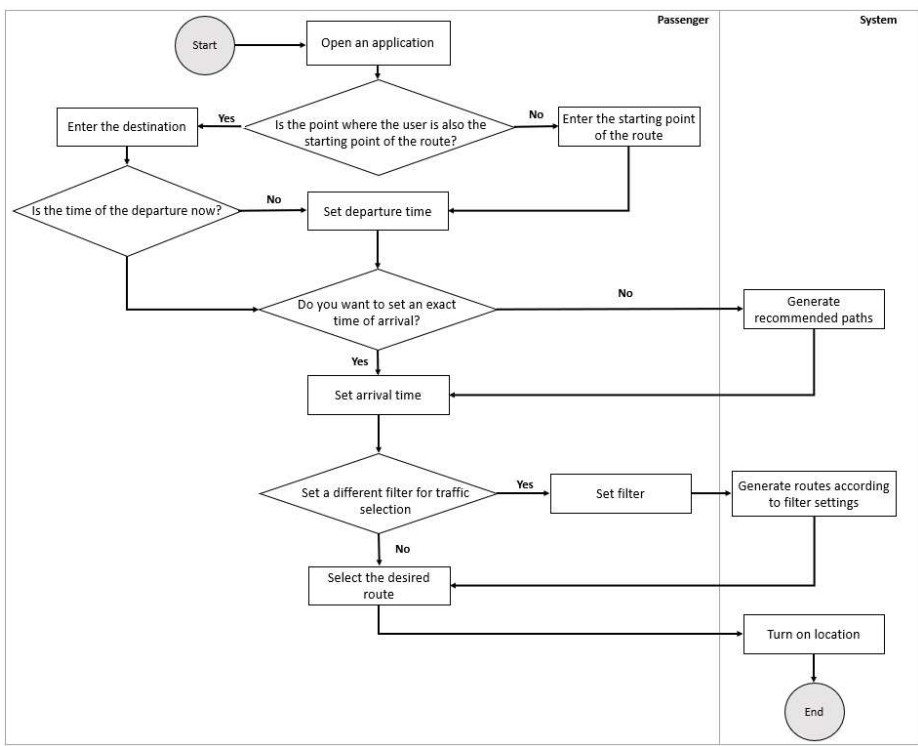

**Figure 2.** Activity diagram of the multimodal application model.

The state diagram was created as a model of the object's behaviour to understand the specification of the sequence of events that the object goes through during its lifetime in response to events. The start is represented by one grey circle and the end by two grey circles. Within the state diagram, nine states were identified (visualised in Figure 3).

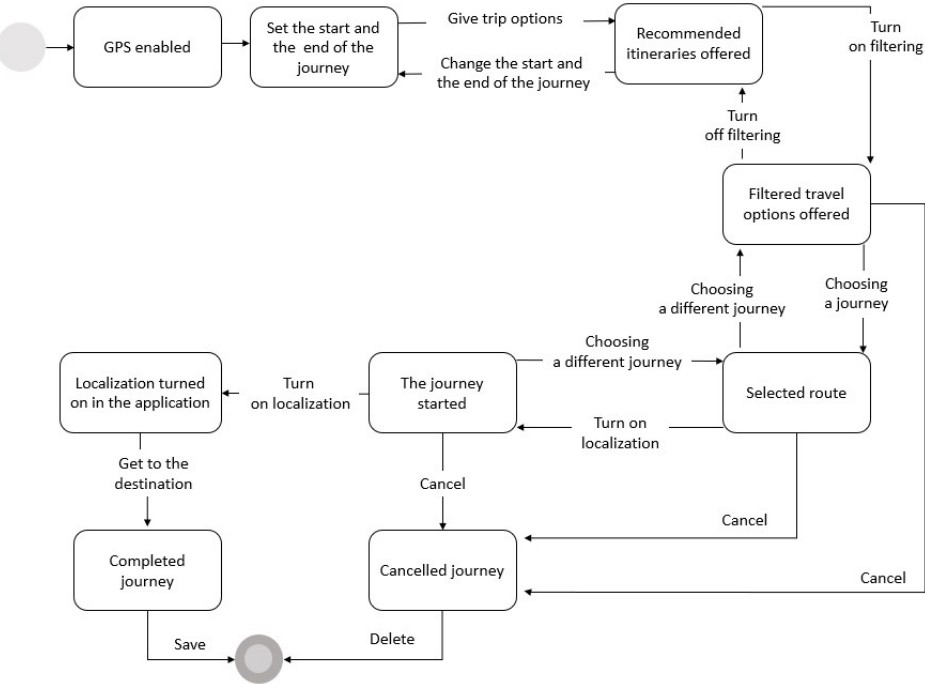

**Figure 3.** Object behaviour state diagram.

The solution provides real-time predictions of traffic levels, analysis of commuting behaviour, and predictions of possible scenarios if a certain type of infrastructure is closed

or altered in any way. With this data, relevant city officials can coordinate the use of city infrastructure to ensure the most efficient traffic flows. The application will allow city officials to promote specific forms of transportation at specific times and create strategies to mitigate traffic. It will also provide travellers with fully integrated navigation solutions that will offer them optimal routes to their parking spaces at any hour. The development and implementation of the application would take six months. The minimum number of registered users has been set at 8110. This number was calculated based on the percentage of the goal concerning the population of the city of Žilina.

### 4.4.2. Creation of Charging Stands for E-Bikes

Another proposal is to create charging stands for e-bikes. Charging infrastructure for e-bikes is crucial to enable cyclists with e-bikes to move around the city efficiently. The city should provide citizens with the ability to charge their e-bikes. Charging stations should be located close to the city centre, in housing estates, and in busy areas where people commute to work. It is also important that charging stations are in safe places under CCTV surveillance, given the financial value of e-bikes. It is recommended to implement five charging stations for e-bikes. The value was calculated based on the ratio of shared e-bikes and charging stations in Stockholm and converted to the proposed number of shared e-bikes in Žilina. Considering the number of bicycles in the city of Žilina (145), the city should purchase seven e-bikes and make them available to residents.

## 5. Discussion

When implementing the proposed solutions for a selected city of a defined scope, adherence to a predefined plan is a pivotal element. For all the benefits that smart cities offer, there are also challenges to overcome. There is a need for the private and public sectors to align with the residents themselves. In the area of implementing electromobility into micro-mobility at the smart city level, the implementation of physical parking stands has a positive impact [77]. Convenience and proximity factors play an important role in promoting good parking behaviour [78]. People tend to park responsibly when reserved parking spaces are available [77]. On the other hand, the dangerous aspects of e-scooter usage are increasingly coming to the fore, resulting in their banning and phasing out of public shared e-scooter systems in cities (e.g., Paris or the UK). It is therefore appropriate to rethink the regulation of e-scooter driving [79].

Bus transport plays a key role in the public transport system and serves as a fundamental pillar of efficient multimodal transport [80]. Buses are the preferred choice for commuting for most people [81]. As governments seek to increase bus ridership, operators are working tirelessly to improve public perception and use of bus services [82]. Providing accurate details of actual departure and arrival times allows passengers to plan their journeys more efficiently while assisting transit agencies in maintaining or improving service performance [83]. Riders who use public transit often rely on multiple buses or trains along with a variety of other modes for last-mile trips [84]. However, considerable time is often wasted in procuring tickets for each part of the journey and during transfers between vehicles. To address this issue, both smart cities and transport operators are actively developing multimodal travel solutions to ensure a seamless experience and promote sustainability [85]. The solution, in this case, is a unified application, connected to a smart city dashboard, which provides real-time traffic information. Additionally, the ecological aspect is supported this way [86,87].

The numerous benefits associated with cycling, including its eco-friendliness, affordability, health benefits, and efficiency in congested areas, spurred government initiatives to promote it, leading to the global spread of bike-sharing schemes [88]. In general, shared electric bicycles lead to a reduction in carbon emissions of 108–120 g per km [86]. In the last five years, electric bicycles have seen a significant increase in popularity. Their ability to provide an environmentally friendly and economical replacement for conventional bicycles and cars has attracted the attention of urban commuters, recreational cyclists, and fitness

enthusiasts. The global electric bicycle market is expected to grow at a CAGR of 9% between 2020 and 2027 [89]. One of the most important elements of the electric bicycle ecosystem is charging stations [90]. With the growing demand for electric bicycles, the need for efficient and affordable charging infrastructure has become important [91]. EV charging station infrastructure should be strategically located near popular cycling routes, urban nodes, and recreational areas [92]. EV charging stations lay the foundations for sustainable transport options, reduce carbon footprint, and promote cleaner cities [30,93].

To ensure a successful transition to sustainable mobility with electromobility features at the smart city level, it is essential to direct the construction of public transport infrastructure towards environmentally responsible roads [94]. To accommodate urban growth and the demand for convenient and accessible mobility, it is essential to maximise the potential of public transport systems in cities [93,95]. Creating an attractive public transport network that integrates seamlessly with walking, cycling, and other forms of micro-mobility is a key element in the pursuit of environmentally sustainable and efficiently managed smart urban areas. The implementation of smart city-level electromobility elements in public transport infrastructure aims to strengthen public transport by optimising and expanding the infrastructure by creating shorter travel times, more capacity, greater system reliability in the event of service disruptions, and more attractive stations and stops.

Planning and approval processes for transport infrastructure projects are subject to lengthy bureaucracy, mainly due to the complex nature of the projects and the number of regulations that need to be complied with [18,96,97]. Delays represent a significant barrier to achieving the goals of expanding public transport and, consequently, to achieving more effective climate protection [18]. There is a need, including leadership and trust, to inform and motivate all stakeholders to actively participate in the progressive development of the system [98]. Advanced transport systems are instrumental in analysing urban mobility patterns, optimising transport performance, and ultimately providing better transport services within the development of a smart city [99,100]. Human involvement in the urban system represents a source of data, occasionally acting as an action member within services, and taking on a decision-making role. It is essential to protect their privacy, security, and well-being [101,102].

Smart city applications can speed up city operations and improve the quality of life for residents. They also enable cities to find and create new value from their existing infrastructure, thereby promoting sustainability [103,104].

## 6. Conclusions

The aim of this article was to look for an answer to the following research question: what role do the smart city strategy and its overall planning play in promoting city sustainability via elements of electromobility in public transport? Therefore, the main objective of the article was to propose recommendations for improving public transport infrastructure in the city of Žilina and similar cities. This was supposed to be carried out based on the analysis of theoretical knowledge and other analyses performed on the selected data. Additionally, the focus needed to be on the elements of electromobility within the framework of smart city standards to support sustainability.

The novelty of this article had three parts. These included the intersection of the described areas (electromobility, smart city, sustainability); comparison and summarisation of best practices (from Singapore, Zürich, Stockholm); and in-depth analysis of the selected city (Žilina).

The knowledge gap stemmed from the fact that compared and summarised studies by world authors mostly focused on only one of the three selected areas. Although they pointed to the many benefits of electromobility in public transport for achieving the sustainability of smart cities, they did not pay enough attention to the strategy and planning considering this topic. The literature also did not provide sufficiently described specific points applicable in cities of comparable size to the city of Žilina.

Smart cities should create an urban environment that delivers a high quality of life for its citizens while generating overall economic growth. Therefore, a top priority of smart cities is their ability to facilitate the delivery of services. In the case of public transport, based on the conclusions of the analyses, cities should focus on making it more attractive and simpler. The benefits of public transport need to be presented to the citizens so that they are willing to trade it for the convenience of a private car. In addition, the analyses show that public transport must be an efficient system for the citizens, enabling them to travel quickly and easily through the city, without having to find parking or being stuck in traffic congestion. It is also important to emphasise the financial aspect and to present the citizens with a model of low-cost transport in the city.

Within the research hypotheses, all three of them (H1: by implementing electromobility elements in the city's public transport infrastructure, the sustainability of the city will be increased; H2: in the case of the city of Žilina, there is no strategy for the implementation of the smart city concept yet; H3: the achievement of sustainability in public transport with electromobility elements must be supported by careful planning) were confirmed. The confirmation of the hypotheses brought the answer to the research question. Thus, the strategy and thorough planning in connection with electromobility elements in public transport play a pivotal role in the effort to achieve sustainability in smart cities.

One of the most important practical benefits of this article is the possibility of applying the conclusions of the analyses performed to the environment of other cities with the number of residents between 50,000 and 100,000, which represents 312 cities in Europe.

The requirements that cities should follow to support the development of an efficient smart city with electromobility elements emerged from case studies, the analysis of the city of Žilina, and the interview with its mayor. The identified requirements include:

- The willingness of cities to incorporate the smart city concept into their agendas is not only an opportunity to improve planning and public service delivery but also an opportunity to modernise the city and revitalise the economy;
- The support at the city government level;
- The support and involvement of city representatives;
- The implementation of citizens' participation;
- Promoting the use of open data from citizens;
- Fostering cooperation between the public and private sectors;
- Embracing transparency and making data accessible to citizens;
- The creation of a smart citizen who is engaged and empowered to contribute positively to the city and community;
- Continuous maintenance and testing of implemented elements to ensure that they work properly;
- Opportunities to monitor energy consumption;
- A solid ecosystem of private sector, academia, society, and organisations that supports the smart city vision;
- A developed connectivity roadmap for IoT devices.

For policymakers and other stakeholders, preferably in the cities identified as medium-sized, the following recommendations were designed. These are linked to the implementation of electromobility elements in public transport infrastructure in the context of a sustainable smart city concept:

- The creation of an initial indicative analysis of the selected area;
- Communication and cooperation with stakeholders;
- Development of a strategy and a plan for its implementation;
- Definition of a clear smart city vision as the basis for coordination of city initiatives;
- Definition of specific business models for each project to achieve sustainability;
- Maintaining transparency.

Future research can focus on other areas of management within the addressed issue. Pertaining to the practical area, further research can be oriented on the following: (1)

summarising knowledge from the areas of public transport infrastructure, electromobility, and smart city and their use for city development; (2) possibilities of applying conclusions from the studied issue to concrete examples; (3) possibilities of using the resulting data from the analysed areas as a secondary source for the city and academia; (4) demonstrating the need to apply management science to the analysed areas; (5) identification of potential risks jeopardising application of the proposed solutions and parallel definition of preventive measures in the case of risks' occurrence; (6) possibilities of applying the conclusions from the examined case studies for cities with a population ranging from 50,000 to 100,000 citizens; (7) expansion of the number of analysed cases; and (8) focusing on different types of cities.

Within the context of this article, it is also important to consider its limitations. To overcome them, in future research activities, researchers could potentially use a variety of resources to improve the accuracy and reliability of response measurements. Focusing on the analysis of three case studies and the analysis of one selected city in this article represents the main limitation. This selection brings the potential for bias and hinders the generalisability of the findings. The focus on medium-sized cities represents the second limitation that can be addressed in future research by, e.g., focusing on small or large cities. Therefore, the limitations mainly followed the data we used in our research. Using other cases as sources of secondary data is always problematic because the cases were originally described for slightly different purposes. In addition, our in-depth analysis only included the interview response from the highest city representative. Adding interviews with other city officials in future research projects can lead to a more precise image of the actual situation being obtained.

**Author Contributions:** Conceptualisation, G.K. and R.J.; methodology, S.K.; software, D.T. and S.K.; validation, D.T. and S.K.; formal analysis, D.T.; investigation, R.J.; resources, D.T.; data curation, S.K.; writing—original draft preparation, G.K., D.T., R.J. and S.K.; writing—review and editing, D.T. and S.K.; visualisation, S.K.; supervision, G.K. and R.J.; project administration, G.K.; funding acquisition, G.K. All authors have read and agreed to the published version of the manuscript.

**Funding:** This research received no external funding.

**Institutional Review Board Statement:** Not applicable.

**Informed Consent Statement:** Informed consent was obtained from all subjects involved in the study.

**Data Availability Statement:** The data of this study are available from the corresponding author upon request.

**Conflicts of Interest:** The authors declare no conflicts of interest. The funders had no role in the design of the study; in the collection, analyses, or interpretation of data; in the writing of the manuscript; or in the decision to publish the results.

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
