# Peer review of "Public Transport Infrastructure with Electromobility Elements at the Smart City Level to Support Sustainability"

_sustainability, doi:10.3390/su16031091_

Round 1

Reviewer 1 Report

Comments and Suggestions for Authors

The paper falls within the scope of the journal and represents a good idea for public transport. However, the first and third sections of this manuscript are graded as preliminary in its current form. Therefore, the paper should be improved by the following comments:

1. The abstract is not written well. Abstract has five-section and you should follow the best practices in your area! I encourage you to mention a summary of your introduction, problem statement, novelty, methodology, and findings in the abstract section. In this version of the abstract, you focused more on the general concepts.

2.     Please bring clear explanations regarding the novelty and main contributions of this research in the introduction section.

3. Please add a paragraph at the end of the Introduction section as the "rest of the paper". You can see a similar paragraph in most of the scientific papers.

4. It is not a good idea to bolt the text in some paragraphs. There are a lot of bolt contexts in the literature review section. Please correct them.

5. The motivation and literature on public transportation should enriched with the following studies:

"Fuzzy ZE-numbers framework in group decision-making using the BCM and CoCoSo to address sustainable urban transportation".

"Providing climate change resilient land-use transport projects with green finance using Z extended numbers based decision-making model."

6. Please add a comparative Table for studies at the end of the literature review section. The literature review is too long and hard to follow up. It's better to add a general Table to mention the summary of the studies and their comparisons.

7. Please add the research gap after adding a literature review Table for the literature review.

8. The material and method section is not clear. Please add more detail to this section.

9. The Discussion and Conclusion sections are good. 

Author Response

Dear reviewer,

thank you very much for your time and expertise you dedicated to the review of our article. We have carefully checked all your remarks and suggestions. We found them very useful and valuable for the increase of the overall quality of the article. We have incorporated them in the revision to the fullest extent.

  1. We have rewritten the whole abstract. We have added the summary of the introduction, problem statement (represented by the research question), description of the article’s novelty, methodology, and he main findings.
  2. We have added a whole new paragraph at the end of the introduction section that describes the article’s novelty in detail. (lines 60-66)
  3. We have added another paragraph with a short summary of the “rest of the paper” at the end of the introduction section. (lines 67-77)
  4. We have eliminated all bolt highlights in the text.
  5. Our motivation for studying the field of public transport is now clearer from the rewritten abstract and introduction section. We have added both suggested references.
  6. The whole literature section was rewritten to improve its flow and clarity for reading. A new comparative table was added at the end of the literature section. (line 222)
  7. The gap identified in the literature that we reviewed for this article was defined in a new paragraph in the literature section. (lines 224-229)
  8. We have made changes in the writing of the methodology section to improve its clarity. We have added the research question in the beginning, which explains the issue our article is addressing. We have also added a new table (line 286) that shows the operationalization of our hypotheses via the indicators set for their testing. All the hypotheses are also connected back to the research question. This adds another level of detail so that the reader can understand our research approach and the overall procedure even better.
  9. We are happy that you liked our discussion and conclusion sections. The changes we had to incorporate in the conclusion were made based on the suggestions and requirements of other reviewers.

We hope that our changes will satisfy your requests and we truly believe that after this revision the overall quality of our article has increased significantly. We also hope that you will be able to recommend the article for publishing in this form.

Have a lovely day.

Authors

Reviewer 2 Report

Comments and Suggestions for Authors

1. In the fourth section, the author introduces multiple cases to support the study's assertions. Nevertheless, a conspicuous deficiency is observed in the absence of a systematically simplified and generalized framework, contributing to the apparent disorganization of this section. A revision and streamlining of this portion are strongly recommended to enhance clarity.

2. It is advisable to organize the content within the figures in a systematic discourse to facilitate knowledge sharing.

3. The introduction and research conclusions should be consistent, ensuring mutual support and coherence.

4. The research conclusions require further validation to ensure they are based on the research question.

5. The primary purpose of the paper is to clearly provide readers with learning opportunities.

6. The relevant discussion before the conclusion should be more targeted, avoiding direct conclusions. Especially, presenting conclusions without any explanation after displaying charts may give the impression that the study is incomplete.

Author Response

Dear reviewer,

thank you very much for the time and expertise you used for your review of our article. We have carefully checked all your suggestions. We found them valuable for the increase in the overall article’s quality. We have incorporated them in the revision to the fullest extent possible.

  1. The whole results section was rewritten to increase its clarity and make it easier to understand by the reader. We have created a new scheme (Figure 1) explaining the levels of our research results and their connection with the hypotheses. This also contributed to increased clarity. Our presentation of the results is now done in a systematic manner. The scheme we added represents a generalized framework underlining our research.
  2. We were not able to change the form of figures 2 and 3 since this is predefined by the specific types of models these diagrams represent. However, we worked with the text around these diagrams where we put additional information about the diagrams and their meaning.
  3. We have made numerous changes both in the introduction and the conclusion section. We have added the description of the article’s novelty and the research gap. We have also increased the coherence of these two sections, as well as of the whole article, via the addition of the research question and the answer to it (the formulation of the research question is in the methodology and its answer is in the conclusion).
  4. We have also added a new table (line 286) that shows the operationalization of our hypotheses via the indicators set for their testing. All the hypotheses are also connected back to the research question that was added as explained in the previous point. This adds another level of detail so that the reader can understand our research approach and the overall procedure even better. Two new tables (Table 4 and Table 6) were added in the results. They show the steps behind the validation of the hypotheses, which are now connected to the research question. All these changes, plus the rewriting of the conclusion section, support the validity of our concluding points.
  5. We respect the perspective where the most important purpose of the article should be the space for learning provided for the readers. We believe that this was strengthened in the article via all the changes we have made, and the information is now more accessible to the readers, enhancing their learning experience.
  6. We have corrected the parts of the article that might have led to its perception as incomplete. This was done by adding explanatory text after figures 2 and 3 (as it was mentioned in detail in point 2 of this response).

We hope that our changes will satisfy your requests and we believe that after this revision the overall quality of our article has increased considerably. We also hope that you will be able to recommend the article for publishing in its present form.

Have a great day.

Authors

Reviewer 3 Report

Comments and Suggestions for Authors

The previous and present theoretical background of the research carried out is adequately described and contextualized. All references cited are relevant to the research.

What is the theoretical or technical support for the construction of the study methodology. A clearer description of the study methodology, research questions and hypotheses is suggested. How the proposed study methodology has been validated.

The results section is difficult to understand. Include Verification or Refutation of Hypotheses, this should be within the results section. Delve into the results in the context of the existing literature on the topic.

The conclusions should adequately follow the main findings of the research. Highlight the main conclusion derived from your work. Indicate theoretical and practical recommendations. It is crucial to discuss methodological drawbacks and potential data problems. Suggest possible avenues for future research and raise questions that could be addressed in subsequent articles.

The manuscript has potential but requires several major revisions. I believe that with these adjustments the article will be much more valuable to the academic community.

Author Response

Dear reviewer,

thank you very much for the time and expertise you used for your review of our article. We have carefully checked all your suggestions. We found them valuable for the increase in the overall article’s quality. We have incorporated them in the revision to the fullest extent possible.

We are happy that you were able to find our theoretical background, the conceptualization of the research as well as the references used adequate. We only made changes that increased the overall clarity, such as a summary table at the end of the literature review section.

The whole methodology section was altered. We have added the research question as well as a table presenting the connection of the research question with the hypotheses and indicators needed for the testing of their validity. The appropriateness of our methodological approach was supported by the addition of several new references to the studies using the same research procedure.

The whole results section was rewritten to make it easier to understand. The understanding is now also supported by the introductory scheme put at the beginning of this section. We have added tables 4 and 6 that are connected to the table with the research indicators presented in the methodology. This supports the verification of our hypotheses.

The whole conclusions section was rewritten. It is now more coherent with the introduction section, showing the novelty and research gap of the article. The points in the conclusions formulated as recommendations are now clearer. They represent the practical implications focused on the managers and policymakers. The methodological drawbacks related to the data and methods used in the research were explained. The possibilities for future research projects were outlined.

Thus, we have made major revisions to the whole text. We hope that our changes will satisfy your requests and we believe that after this revision the overall quality of our article has increased considerably. We also hope that you will be able to recommend the article for publishing in its present form.

Have a great day.

Authors

Reviewer 4 Report

Comments and Suggestions for Authors

Dear Authors,

Your paper is interesting; however, it requires a few changes.

In the text, reference numbers should be placed in square brackets [ ], and placed before the punctuation; for example [1], [1–3] or [1,3]. For embedded citations in the text with pagination, use both parentheses and brackets to indicate the reference number and page numbers; for example [5] (p. 10). or [6] (pp. 101–105).

Lines 46-47: "For millennials, public transportation serves as the preferred way to promote community connections and engage in digital socialization" It is intriguing; however, could you provide the research supporting this claim? Can you please cite the source?

Lines: 48-49 "The American Public Transportation Association (APTA) claims that public transportation is ten times safer per mile compared to other modes of transportation". The source says: by any other means of individual transport such as a car.

Lines 95-96: "From the passengers’ perspective, the fact that an electric or diesel bus will drive to a stop does not determine their transport decisions"  On what research is this claim based? Could you provide the source?

Line 150: "we decided to investigate its application in a selected city" You selected the City of Žilina; could you explain why? Please add this information to the text.

Lines 526-527: "The number of people transported by public transport (public transport) in 2021 was 16,806 (Annual Report, 2021)". Did you mean 16 806 000?

Lines 536-537: "bicycles end up at the Žilina Waterworks, where people use 536
them for recreation and then do not return them to the city centre".So, how do people return to the city center? Do they use buses?

Lines 638-640: "Based on the analyses performed, a solution for the selected City of Žilina was defined to increase the number of passengers using public transport and alternative forms of transport in the City of Žilina by 10%".  Can you provide the reasoning behind deciding on a 10% increase? It's unclear why 10% was chosen instead of, for example, 15%.

Lines 673-674: "The minimum number of registered users has been set at 24,158". Why? How did you arrive at this value?

Lines 722-723: "Recently, electric bicycles have seen a significant increase in
popularity". hat does 'recently' mean? For example, in the last 2 years or in the last two days?

Author Response

Dear reviewer,

thank you very much for the time and expertise you used for your review of our article. We have carefully checked all your suggestions. We found them valuable for the increase in the overall article’s quality. We have incorporated them in the revision to the fullest extent possible.

Thus far, we have been working with the style of referencing with round brackets because it is easier to add new references to the text as requested during the review process. Once we get approval for final changes and proofreading of the article to be published in relation to its content, we will change the style of referencing in accordance with the journal’s standards.

We have added references to all the statements you included in your review as those that were not sufficiently supported by sources of information. All the changes made in the text are highlighted in yellow colour.

We have expanded our explanation for selecting the City of Žilina for our research.

The number of people transported by public transport in the city was corrected. Thank you very much for spotting this mistake and helping us eliminate it in the article.

We have added an explanation of why the shared bicycles can end up at the waterworks site.

We have extended the explanation for setting the target value at 10% - this was based on the research results, calculations using the total number of citizens, as well as the realistic aspect of the goal set this way.

The minimum number of registered users was also explained in more detail in the highlighted changes.

We have added a more specific timeframe for the observation of the increase in electric bicycles' popularity.

We hope that our changes will satisfy your requests and we believe that after this revision the overall quality of our article has increased considerably. We also hope that you will be able to recommend the article for publishing in its present form.

Have a great day.

Authors

Round 2

Reviewer 2 Report

Comments and Suggestions for Authors

The paper has been revised adequately. Well done, authors!

Reviewer 4 Report

Comments and Suggestions for Authors

The manuscript has been improved. So I can accept it.